# Stress Reduction Effects during Block-Tapping Task of Jaw in Healthy Participants: Functional Near-Infrared Spectroscopy (fNIRS) Measurements of Prefrontal Cortex Activity

**DOI:** 10.3390/brainsci12121711

**Published:** 2022-12-14

**Authors:** Takahiro Kishimoto, Takaharu Goto, Tetsuo Ichikawa

**Affiliations:** 1Department of Comprehensive Dentistry, Graduate School of Biomedical Sciences, Tokushima University, Tokushima 770-8504, Japan; 2Department of Prosthodontics and Oral Rehabilitation, Graduate School of Biomedical Sciences, Tokushima University, Tokushima 770-8504, Japan

**Keywords:** prefrontal cortex, functional near-infrared spectroscopy, stress reduction, periodontal ligament, chewing

## Abstract

The purpose of this study was to explore the influence of simple jaw opening and closing task of chewing movements on the activities of the prefrontal cortex (PFC) from the viewpoint of stress reduction. We measured cerebral blood flow (CBF) in the PFC during a block-tapping task of the jaw in healthy participants. Eleven young healthy individuals with normal dentition (7 males and 4 females, mean age 28.0 ± 3.7 years) volunteered for this study. CBF was measured using a wearable, functional near-infrared spectroscopy device. Measurements were taken using the central incisors and first molars at interocclusal distances of 5 and 10 mm. The participants were asked to bite a hard/soft block. CBF in all conditions showed limited variability or decreasing trend compared to resting state before the task. The main effect was observed for interocclusal distance (*p* = 0.008), and there were no significant differences for measurement area of the PFC, tooth type, and material type. An interaction was found between tooth type and material type. In conclusion, these results suggest that simple and rhythmical chewing motion has an effect of reducing CBF in the PFC and resting the PFC, which is an especially notable aspect of periodontal sensory information in the molar.

## 1. Introduction

The prefrontal cortex (PFC) is located in the anterior region of the cerebral cortex [1,2]. The PFC plays a role in information processing of higher-order behavior, such as complex planning, working memory, attention, and decision-making [3,4,5,6]. In recent years, an increasingly elderly population has led to an exponential growth in the number of patients with dementia and mild cognitive impairment [7]. Thus, PFC function, which underlies cognitive functioning, has received more attention lately [8,9]. Several studies have reported the relationship between cognitive functions and motor tasks [10,11,12]. In particular, reports have noted that poor masticatory function due to tooth loss could increase the risk of cognitive function decline [13,14,15].

An increase in cerebral blood flow (CBF) during the task of maintaining occlusal force, which needs higher brain function such as working memory and much attention, was reported in our previous study [16]. Moreover, an increase in CBF was observed when the periodontal sensory information in the molar was primarily responsible for controlling the occlusal force, and it was suggested that the PFC contributes to the inhibition of excessive occlusal force executed by the teeth.

On the other hand, many studies have reported that gum chewing contributes to relieving stress and anxiety. Scholey et al. employed cognitive tasks as laboratory stressors and reported that gum chewing had the effect of reducing salivary cortisol levels as a physiological marker of stress and improving task performance compared to non-chewing condition [17]. Several studies investigating the effects of chewing on salivary cortisol have shown that cortisol level decreases with stronger chewing force or faster chewing rate [18,19].

Regarding the relationship between stress and PFC, Yu et al. used functional magnetic resonance imaging to show that gum chewing relieved stress by both attenuating the sensory processing of noise-induced stress and inhibiting the propagation of stress-related information in the brain stress network [20]. Therefore, it has been suggested that gum chewing could relieve stress. In these studies, although stress levels were evaluated using the blood pressure, heart rate, electroencephalography, and stress markers in the saliva, such as cortisol or IgA level, they were measured only just before and after the tasks. There have been no studies assessing how CBF in the PFC changes throughout the chewing movement.

The purpose of this study was to explore the influence of simple jaw opening and closing task of chewing movements on the activities of the PFC from the viewpoint of stress reduction. We measured CBF of the PFC during a block-tapping task, which involved biting several types of blocks of different thickness and hardness by incisors or molars in a constant rhythm, and evaluated the effect on stress reduction.

## 2. Materials and Methods

### 2.1. Participants

Eleven young healthy individuals with normal dentition (7 males and 4 females, mean age 28.0 ± 3.7 years) and without any stomatognathic or central nervous system disorders volunteered for this study. This study was conducted with the approval of the Ethics Committee of the Tokushima University Hospital (No.1780). All experiments were performed according to the approved guidelines. The experimental protocol was explained to participants, and informed consent was obtained.

### 2.2. Measurement of Prefrontal Cortex Activity

PFC activity was assessed by CBF in unrestrained participants using a wearable, functional near-infrared spectroscopy device (WOT-100, NeU, Tokyo, Japan). This device was used to measure relative changes in the concentration of oxyhemoglobin via light attenuation at two wavelengths (705 and 830 nm) that easily passed through skin, tissue, and bone at 5 Hz. This device had a total of 22 measuring probes, out of which 10 were spaced at an interval of 30 mm that covered three measurement areas located around the frontal pole. Probes 10, 11, 12, and 13 were for Brodmann’s area 10 of the dorsolateral PFC; probes 7, 8, and 9 were for the right side of area 46; and probes 14, 15, and 16 were for the left side of area 46. Initially, CBF signals were qualitatively assessed. Then, the changes in CBF at task completion were calculated based on the CBF just before task onset.

### 2.3. Experimental Task

The block-tapping task is a simple repeated jaw opening and closing movement onto a block (Figure 1). Hard (acrylic resin; UNIFAST II, GC Corporation, Tokyo, Japan) and soft (silicone rubber; EXADENTURE, GC Corporation, Tokyo, Japan) blocks were prepared to mimic hard and soft foods, respectively, with two dimensions (10 mm × 10 mm × 5 mm and 10 mm × 10 mm × 10 mm).

The block was held by left-side central incisors or first molars through a special handle jig. Participants were asked to bite each block of material at a regular interval (1 Hz) for a total of 30 s at each interocclusal distance (IOD) as a simple jaw opening and closing movement onto a block. Eight conditions were defined from the different tooth types (incisors/molars), material types (hard/soft), and IODs of 5 and 10 mm referring to two heights of block. Each condition was randomly performed four times for 30 s with a rest interval of 60 s and over the course of separate days (maximum interval: 14 days) for each participant.

### 2.4. Statistical Analysis

Four-way analysis of variance with Tukey post hoc was used to compare the factors of tooth type, IOD, material type, and measurement area. Unpaired *t*-test was used to compare between two groups. Statistical analysis was performed at a significance level of 0.05 with SPSS^®^ version 25.0 software (IBM Corp., Armonk, NY, USA).

## 3. Results

Figure 2 shows the signal patterns of CBF in the PFC for 30 s from task onset to the end using the average of the participants’ signals at the task onset. The CBF patterns in almost all conditions showed limited variability compared to resting state before the task and showed a decreasing trend for the conditions of incisor/hard and molar/soft with IOD of 5 mm. This result was different from that reported in our previous study on maintaining occlusal force task.

Figure 3 shows the difference in CBF just before the task and at the completion of the task. According to the results of the four-way analysis of variance with Tukey post hoc, the main effect was observed for IOD (*p* = 0.008), and there were no significant differences for measurement area of the PFC, tooth type, or material type. An interaction was found between tooth type and material type (*p* = 0.016). Regarding the difference between IODs of 5 and 10 mm, the change in the CBF for IOD of 5 mm was significantly lower than that for 10 mm. Regarding the difference in tooth type and material type, the CBF when biting the soft block was significantly lower than when biting the hard block, and the result was observed only for biting by molars.

## 4. Discussion

We evaluated CBF of the PFC during block-tapping of jaw, which is a simple task that occurs during activities such as gum chewing.

It has been reported that several loops relating to control and learning of movement interconnect cerebral cortical areas. This includes the prefrontal loop, which outputs signals from the PFC via the caudate nucleus and the thalamus to the motor unit and projects them back to the PFC, and the motor loop, which outputs signals from the supplementary motor cortex (SMC) via the putamen and the thalamus and projects them back to the SMC without the PFC [21,22,23]. In our previous study, using the same experimental apparatuses, we reported that CBF in the PFC increased during maintaining occlusal force task, which requires participants to understand the motion and to learn quantitative and temporal force control, as PFC processes increased due to periodontal tactile input from molars [16]. Judging from these results, block-tapping task could be regulated by the motor loop without supervisory attention from the PFC because the task is characterized by simple and repetitive jaw open/close movements and based on already acquired motion sequence habits.

The decrease in activity in the PFC observed during the simple block-tapping task, especially in healthy young participant without cognitive decline, may be regarded as relaxation considering the stress reduction effects of gum chewing reported in previous studies [20,24,25]. Moreover, the CBF decreased when biting a soft block by molars, showing the interaction between tooth type and material type. The condition is similar to rhythmical gum-chewing motion, which grinds food and further softens it. The result seems to be the same as the stress reduction effect induced by gum chewing reported in previous studies.

The CBF measured for IOD of 5 mm was significantly less than that for IOD of 10 mm. The results could be explained by the difference in primary mechanoreceptors responsible for adjusting occlusal force for the two different IODs. Manly et al. reported that periodontal sensory receptors primarily dominate the controlling occlusal force when the IOD is less than 5 mm [26]. On the other hand, Christensen et al. reported that somatic sensory information from the temporomandibular joint and the muscle spindles of the masseters are primarily responsible for adjusting the occlusal force when the IOD is greater than 10 mm [27]. Therefore, it is likely that sensory information is processed via the prefrontal loop when sensory information is transmitted from more kinds of sensory receptors.

The sample size should be considered as a limitation of this study. In this study, setting the alpha value of 0.05, the power of 0.80, and the effect size of moderate would require the sample size of about 60 subjects. Very few of the previous studies targeting CBF have used a sample size of about 60 people. Thus, the results of this study, obtained from a sample size of 11 subjects, is thought to be significant. However, it is possible that a larger sample size may make the results obtained in this study more pronounced.

As for maintaining occlusal force task, which is a complicated and unskilled task, the increase in CBF in the PFC was observed only when the task was performed using molars. In contrast, for block-tapping, which is a simple and skilled task, it decreased only with molars. These results suggest that simple and rhythmical chewing motion has an effect of reducing CBF in the PFC and resting the PFC, which is an especially notable aspect of periodontal sensory information in the molar.

## Figures and Tables

**Figure 1 brainsci-12-01711-f001:**
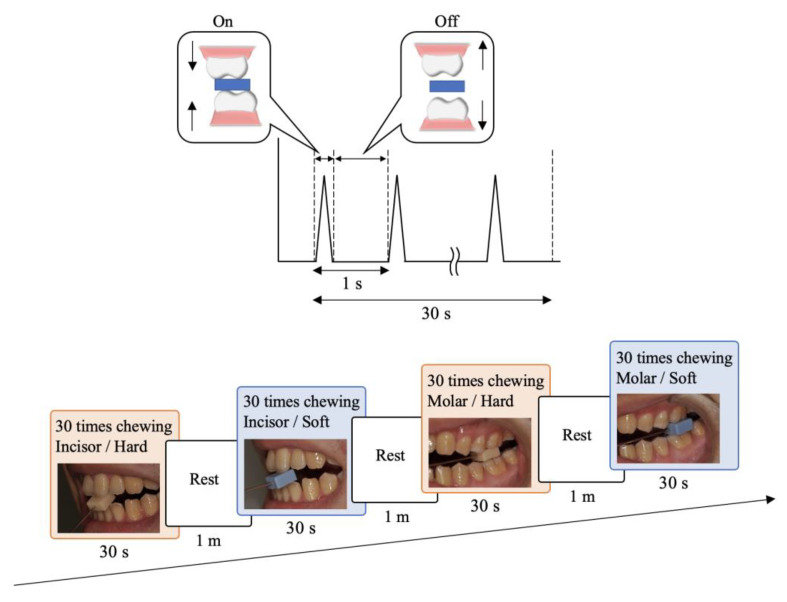
Serial measurement of each condition. The order of experimental tasks was randomized and repeated four times.

**Figure 2 brainsci-12-01711-f002:**
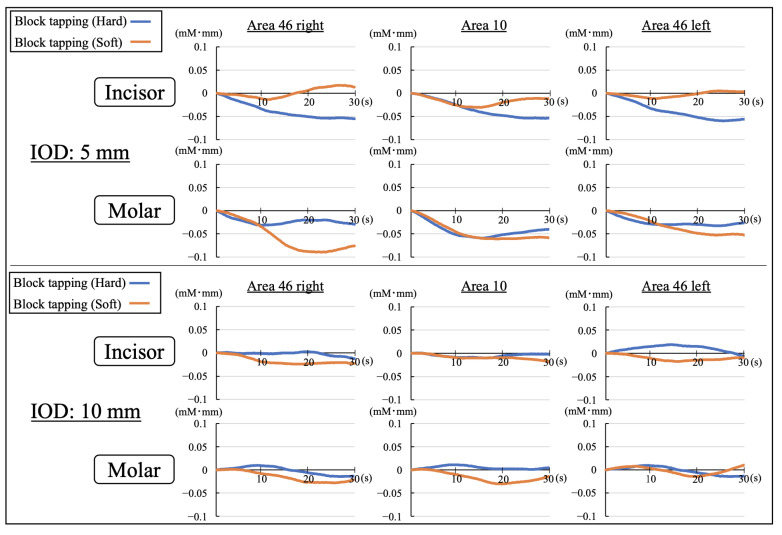
Signal averaging patterns of the cerebral blood flow in the prefrontal cortex with IODs of 5 and 10 mm (IOD: interocclusal distance).

**Figure 3 brainsci-12-01711-f003:**
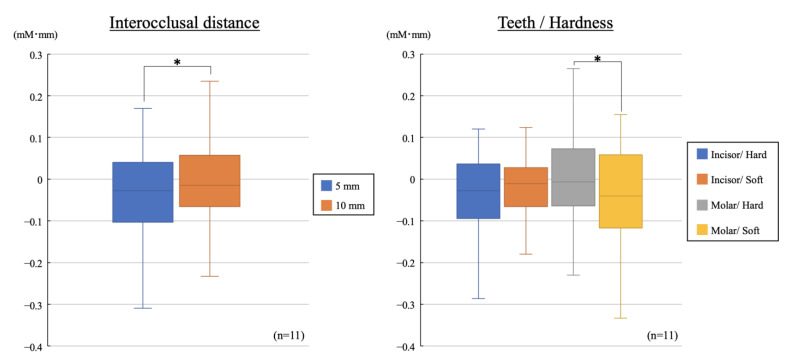
Changes in the cerebral blood flow for different interocclusal distance, teeth type, and block hardness (* *p* < 0.05, unpaired *t*-test).

## Data Availability

The data presented in this study are available on request from the corresponding author.

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
