# Peer review of "Stress Reduction Effects during Block-Tapping Task of Jaw in Healthy Participants: Functional Near-Infrared Spectroscopy (fNIRS) Measurements of Prefrontal Cortex Activity"

_brainsci, 2022, doi:10.3390/brainsci12121711_

Round 1
Reviewer 1 Report
In the present manuscript, the authors study how jaw-opening and closing tasks affect PFC activities. They measured PFC activities via CBF using NIRS and found CBF decreased during the tasks. They suggest a decrease in the PFC activity may be related to a decrease in stress levels since - if I understand correctly - jaw opening and closing accompanied by chewing movements are reported to decrease stress and anxiety. I found the results looked interesting; however, I am confused when I saw the authors’ previous study (Kishimoto et al. Exp. Brain Res. 2019) and would like to know what caused the differences.
1. In the previous study, CBF in the PFC increased during the biting tasks. It seems that the tasks used in the present study also involve occlusal force which increases CBF in the PFC. Nonetheless, CBF only decreased in the current tasks. The authors discuss simple vs. skilled task differences, but why do these differences evoke opposite CBF responses? Could you elaborate on it a little more, please?
2. How did the spatial patterns of CBF in the PFC look like? Were both increases and decreases in CBF observed within a single area as in the previous study (Fig. 2)? If so, what does that mean?
3. Were there any laterality of CBF responses in the PFC for the left or right molar bite tasks?
4. This might be a little off-topic, but what decreases the PFC activity? If local inhibitory neuronal activities caused the decreased PFC activity, it would suggest that the excitation of inhibitory neurons does not increase CBF despite their increased metabolic demand. This may bring some insight into the neurovascular control of inhibitory neurons.
Author Response
Response to the reviewers’ comments:
We wish to express our strong appreciation to the reviewers for their insightful comments on our paper. We have revised according to the reviewers’ suggestions. In addition, we have added the figure for a better understanding of our study results and corrected throughout the text.
- In the previous study, CBF in the PFC increased during the biting tasks. It seems that the tasks used in the present study also involve occlusal force which increases CBF in the PFC. Nonetheless, CBF only decreased in the current tasks. The authors discuss simple vs. skilled task differences, but why do these differences evoke opposite CBF responses? Could you elaborate on it a little more, please?
Response 1:
We wish to thank the reviewer for this comment. The task used in our previous study involved maintaining occlusal force within an instructed range, and have not been tested before, that is un-skilled task. Conversely, the block-tapping task was a simple repeated jaw opening and closing movement onto a block without needing a complex control of occlusal force. As mentioned in Discussion section, the loops concerning control and learning of movement interconnectted to cerebral cortical areas is thought to effect on the differences of results obtained from these two tasks. Thus, maintaining occlusal force task would be involved the prefrontal loop because the task was characterized as complex movement with controlloing occlusal force, and block-tapping task would be regulated by the motor loop without supervisory attention from the PFC because the task was characterized as simple and repetitive jaw open-close movements. These differences are thought to effect on our results.
- How did the spatial patterns of CBF in the PFC look like? Were both increases and decreases in CBF observed within a single area as in the previous study (Fig. 2)? If so, what does that mean?
Response 2:
Regarding the topographic patterns at the measurement completion, both red and blue colors, indicated the increase and decrease of CBF, were observed in the most experimental conditions as shown in the figure of our previous study (Fig. 2). However, since the bule/red color were pale, the results were considered to show no change or a slight decreaseing in CBF. Such topographic patterns are common in the assessment of CBF. Moreover, no significant differences were found through probe-by-probe analysis.
- Were there any laterality of CBF responses in the PFC for the left or right molar bite tasks?
Response 3:
In this study, the effect of laterality (right/left side) of CBF responses in the PFC was not investigated because the task was performed on only left side central incisors and first molars. Regarding the measurement area, no significant differences were found between right and left side of area 46 in this result.
- This might be a little off-topic, but what decreases the PFC activity? If local inhibitory neuronal activities caused the decreased PFC activity, it would suggest that the excitation of inhibitory neurons does not increase CBF despite their increased metabolic demand. This may bring some insight into the neurovascular control of inhibitory neurons.
Response 4:
Thank you for your valuable comments. This study was designed to measure CBF in the PFC, and we cannot refer to the activity of neurons. Therefore, mentioned in above response 1, the loops concerning control and learning of movement interconnectted to cerebral cortical areas is thought to effect on the decreasing PFC activity. Thus, maintaining occlusal force task would be involved the prefrontal loop because the task was characterized as complex movement with controlloing occlusal force, and block-tapping task would be regulated by the motor loop without supervisory attention from the PFC because the task was characterized as simple and repetitive jaw open-close movements. These differences are thought to effect on our results.

Reviewer 2 Report
This work compares the cerebral blood flow (CBF) measured from the prefrontal cortex (PFC) during a repetitive block-tapping task with different conditions (soft or hard material; bitting with incisors or molars; bitting with two different interocclusal distances (5 and 10 mm). The study is interesting and original, and the manuscript is well-written. However, several issues need to be addressed, as described below:
(1) Statistical analysis. Both parametric analyses comparing mean values (analysis of variance) and non-parametric analyses comparing median values (Wilcoxon signed-rank test) were used. It is unclear why. Did the authors test the CBF results for normal distribution?
(2) Results section. Figure 3 shows medians, but the results reported from the analysis of variance compare mean values (lines 112 to 115). Moreover, the results in Figure 3 show only differences in median values compared by interocclusal distance (OID) or by teeth and hardness. It is confusing why it mixed the result from a parametric analysis that compares the mean values with the results comparing medians from the separated factors (teeth type and hardness).
(3) Results, lines 107 to 109. The sentence "The CBF patterns in all conditions showed limited variability or decreasing trend compared with those of resting state before task" is unclear. Was the trend measured to classify the change as decreasing trend? Or the "decreasing trend" refers to no statistically significant decreasing change compared to the baseline? If the change from baseline was not significant (i.e., the value of the change during the task was not different from zero), then the authors cannot state that there was a trend when in fact was no significant change or no change.
(4) Figure 3. Please identify the number of measurements included in each result of this figure (sample size).
(5) Line 143. Add a reference to this sentence.
(6) Discussion section. Please add a section about study limitations. The sample size is small. Perhaps larger samples are required to observe significant changes if the compared effects are small.
Author Response
Response to the reviewers’ comments:
We wish to express our strong appreciation to the reviewers for their insightful comments on our paper. We have revised according to the reviewers’ suggestions. In addition, we have added the figure for a better understanding of our study results and corrected throughout the text.
(1) Statistical analysis. Both parametric analyses comparing mean values (analysis of variance) and non-parametric analyses comparing median values (Wilcoxon signed-rank test) were used. It is unclear why. Did the authors test the CBF results for normal distribution?
Response 1:
We wish to thank the reviewer for this comment. I apologize for no state about normality of the data. The data were normaly distributed, thus the comparison between two groups has been modified to parametric analysis. We re-analyzed the data and there was no change to the results.
(2) Results section. Figure 3 shows medians, but the results reported from the analysis of variance compare mean values (lines 112 to 115). Moreover, the results in Figure 3 show only differences in median values compared by interocclusal distance (OID) or by teeth and hardness. It is confusing why it mixed the result from a parametric analysis that compares the mean values with the results comparing medians from the separated factors (teeth type and hardness).
Response 2:
As mentioned in Response 1, we re-analyzed the data with parametric test and confirmed that there was no change to the results. The results of the analysis of variance showed main effect for difference in interocclusal distance (IOD) and interaction for differences in teeth type and block hardness, thus the comparisons between two groups have been conducted for differences in above conditions. Figure 3 showed the results that significant differences were found.
(3) Results, lines 107 to 109. The sentence "The CBF patterns in all conditions showed limited variability or decreasing trend compared with those of resting state before task" is unclear. Was the trend measured to classify the change as decreasing trend? Or the "decreasing trend" refers to no statistically significant decreasing change compared to the baseline? If the change from baseline was not significant (i.e., the value of the change during the task was not different from zero), then the authors cannot state that there was a trend when in fact was no significant change or no change.
Response 3:
Although conditions were found in which cerebral blood flow (CBF) at the task completion decreased compared to the baseline, no statistically significant decrease was found in such condition. Although it was not statistically significant, the "decreasing trend" in this section shows the qualitatively observed decrease in the condition where tapping hard blocks on incisors and soft blocks on molars at interocclusal distance (IOD) of 5 mm compared to the other conditions.
The sentence pointed out was not clear and revised as follows.
The CBF patterns in all conditions showed limited variability or decreasing trend compared with those of resting state before task.
↓
(Lines 110 to 113)
The CBF patterns in almost conditions showed limited variability compared with those of resting state before task, and showed decreasing trend in conditons of Incisor/Hard and Molar/Soft with IOD of 5 mm.
(4) Figure 3. Please identify the number of measurements included in each result of this figure (sample size).
Response 4:
Sample size (n=11) have been added in the Figure 3.
(5) Line 143. Add a reference to this sentence.
Response 5:
The following references have been added in line 152.
[20]. Yu, H.; Chen, X.; Liu, J.; Zhou, X. Gum chewing inhibits the sensory processing and the propagation of stress-related infor-mation in a brain network. PLoS ONE 2013, 8, e57111.
[24]. Sasaguri, K.; Otsuka, T.; Tsunashima, H.; Shimazaki, T.; Kubo, KY.; Onozuka, M. Influence of restoration adjustments on prefrontal blood flow: A simplified NIRS preliminary study. Int. J. Stomatol. Occlusion Med. 2015, 8:22-28
[25]. Nagasawa, Y.; Ishida, M.; Komuro, Y.; Ushioda, S.; Hu, L.; Sakatani, K. Relationship Between Cerebral Blood Oxygenation and Electrical Activity During Mental Stress Tasks: Simultaneous Measurements of NIRS and EEG. Adv. Exp. Med. Biol. 2020, 1232, 99-104
(6) Discussion section. Please add a section about study limitations. The sample size is small. Perhaps larger samples are required to observe significant changes if the compared effects are small.
Response 6:
We strongly appreciate the reviewer's comment on this point. The following sentence have been added in Discussion section, lines 165 to 171.
The sample size should be considered as limitation of this study. In this study, setting the alpha value of 0.05, the power of 0.80, and the effect size of moderate would require the sample size of about 60 subjects. In previous studies targeting CBF, such as our study, there were very few studies that were examined with a sample size of about 60 people, thus the results in this study, obtained from the sample size of 11 subjects is thought to be also significant. However, it is possible that a larger sample size may make the results obtained in this study more pronounced results.

Round 2
Reviewer 2 Report
No further comments.